# The Role of Aspartate Aminotransferase-to-Lymphocyte Ratio Index (ALRI) in Predicting Mortality in SARS-CoV-2 Infection

**DOI:** 10.3390/microorganisms11122894

**Published:** 2023-11-30

**Authors:** José Manuel Reyes-Ruiz, Omar García-Hernández, Gustavo Martínez-Mier, Juan Fidel Osuna-Ramos, Luis Adrián De Jesús-González, Carlos Noe Farfan-Morales, Selvin Noé Palacios-Rápalo, Carlos Daniel Cordero-Rivera, Tatiana Ordoñez-Rodríguez, Rosa María del Ángel

**Affiliations:** 1Department of Research, Unidad Médica de Alta Especialidad, Hospital de Especialidades No. 14, Centro Médico Nacional “Adolfo Ruiz Cortines”, Instituto Mexicano del Seguro Social (IMSS), Veracruz 91897, Mexico; gustavo.martinezmi@imss.gob.mx; 2Department of Internal Medicine, Unidad Médica de Alta Especialidad, Hospital de Especialidades No. 14, Centro Médico Nacional “Adolfo Ruiz Cortines”, Instituto Mexicano del Seguro Social (IMSS), Veracruz 91897, Mexico; haro-green@hotmail.com (O.G.-H.); tatiana.ordonez@imss.gob.mx (T.O.-R.); 3Facultad de Medicina, Universidad Autónoma de Sinaloa, Culiacán 80019, Mexico; osunajuanfidel.fm@uas.edu.mx; 4Unidad de Investigación Biomédica de Zacatecas, Instituto Mexicano del Seguro Social, Zacatecas 98000, Mexico; luis.dejesus@cinvestav.mx; 5Departamento de Ciencias Naturales, Universidad Autónoma Metropolitana (UAM), Unidad Cuajimalpa, Mexico City 05348, Mexico; carlos.farfan@cinvestav.mx; 6Department of Infectomics and Molecular Pathogenesis, Center for Research and Advanced Studies (CINVESTAV-IPN), Mexico City 07360, Mexico; selvin.palacios@cinvestav.mx (S.N.P.-R.); carlos.cordero@cinvestav.mx (C.D.C.-R.); rmangel@cinvestav.mx (R.M.d.Á.)

**Keywords:** COVID-19, SARS-CoV-2, aspartate aminotransferase-to-lymphocyte ratio index (ALRI), biomarker, mortality

## Abstract

COVID-19 has a mortality rate exceeding 5.4 million worldwide. The early identification of patients at a high risk of mortality is essential to save their lives. The AST-to-lymphocyte ratio index (ALRI) is a novel biomarker of survival in patients with hepatocellular carcinoma, an organ susceptible to SARS-CoV-2 infection. For this study, the prognostic value of ALRI as a marker of COVID-19 mortality was evaluated. For this purpose, ALRI was compared with the main biomarkers for COVID-19 mortality (neutrophil-to-lymphocyte ratio [NLR], systemic immune-inflammation index [SII], platelet-to-lymphocyte ratio [PLR], lactate dehydrogenase (LDH)/lymphocyte ratio [LDH/LR]). A retrospective cohort of 225 patients with SARS-CoV-2 infection and without chronic liver disease was evaluated. In the non-survival group, the ALRI, NLR, SII, and LDH/LR were significantly higher than in the survival group (*p_corrected_* < 0.05). ALRI had an area under the curve (AUC) of 0.81, a sensitivity of 70.37%, and a specificity of 75%, with a best cut-off value >42.42. COVID-19 patients with high ALRI levels had a mean survival time of 7.8 days. Multivariate Cox regression revealed that ALRI > 42.42 (HR = 2.32, 95% CI: 1.35–3.97; *p_corrected_* = 0.01) was a prognostic factor of COVID-19 mortality. These findings prove that ALRI is an independent predictor of COVID-19 mortality and that it may help identify high-risk subjects with SARS-CoV-2 infection upon admission.

## 1. Introduction

In December 2019, China (Wuhan, Hubei Province) reported cases of pneumonia caused by an unknown virus [1]. Later, on 11 February 2020, the new virus was named Severe Acute Respiratory Syndrome Coronavirus 2 (SARS-CoV-2) by the International Committee on Taxonomy of Viruses (ICTV), and the resulting disease was named COVID-19 by Word Health Organization (WHO) [2]. COVID-19 has characteristics of viral pneumonia that can lead to respiratory failure, respiratory distress syndrome, and, ultimately, death [1]. During SARS-CoV-2 infection, there is an immune response imbalance due to hyperinflammation, leading to a poor prognosis of COVID-19. Thus, due to the high COVID-19 mortality rate [3], it is important to screen all patients with hyperinflammation using hematologic parameters or ratios and to predict disease progression early on to decrease COVID-19 mortality. In this context, identifying biomarkers that can reflect inflammation and immune status are potential predictors for the prognosis of COVID-19 [4]. Inflammation parameters from routine bloodwork, such as leukocytes [5], neutrophiles [6], lymphocytes [6], neutrophil-to-lymphocyte ratio (NLR) [7], platelet-to-lymphocyte ratio (PLR) [7], systemic immune-inflammation index (SII) [8], lactate dehydrogenase (LDH)/lymphocyte ratio [9], aspartate aminotransferase (AST)-to-neutrophil ratio index (ANRI) [10], and AST-to-platelet ratio index (APRI) [11] have been demonstrated to be efficient prognostic biomarkers for COVID-19.

SARS-CoV-2 infection induces alterations in liver function tests and hepatic impairment, according to pathological findings concerning patients with COVID-19 [12]. An increase in AST levels at admission has been identified as an independent predictor of COVID-19 mortality [12]. The AST-to-lymphocyte ratio index (ALRI) is a novel inflammatory index for hepatocellular carcinoma, and it is related to the survival of patients with this condition [13]. A study conducted in a hospital in Asia (Ankara, Turkey) found no relationship between ALRI and COVID-19 severity [11]. However, its role in COVID-19 mortality has yet to be explored. Thus, this study aims to assess the prognostic value of ALRI regarding the mortality of SARS-CoV-2-infected patients upon hospital admission. For this purpose, a comparison of the main biomarkers for COVID-19 mortality (lymphocytes, platelets, NLR, SII, PLR, LDH/lymphocyte ratio) was also performed.

## 2. Materials and Methods

### 2.1. Study Design and Participants

A retrospective single-center observational and cross-sectional study was conducted at a tertiary care hospital from the Mexican Social Security Institute (IMSS): Unidad Médica de Alta Especialidad, Hospital de Especialidades No. 14, Centro Médico Nacional “Adolfo Ruiz Cortines”. This study was approved by our local research ethics committee (R-2023-3001-111) and conducted following the Declaration of Helsinki. Patients without chronic liver disease and with a positive nucleic acid test for SARS-CoV-2 from May to September 2020 were included in the study. This study only included patients with in-hospital mortality due to COVID-19. Patients with other causes of mortality and with additional SARS-CoV-2 infection were excluded from the study. Also, patients <18 years of age, patients missing a baseline value of aspartate aminotransferase (AST) or lymphocytes, and patients with acquired immune deficiency syndrome, a malignant tumor, liver cirrhosis, or pregnancy were excluded.

### 2.2. Data Collection

The demographic data, laboratory results, clinical features, and medical history of the patients included in the study were obtained from electronic medical records. Data collected from laboratory results were defined using the first-time examination at admission (within 24 h after admission). Hematological data were tested in the same laboratory using the same standard. To analyze mortality risk, patients were followed up from admission to discharge (1 to 22 days). The follow-up data were collected through reviewing medical records using a double-blind method.

### 2.3. Variables

#### 2.3.1. Outcome Variable: Mortality

The in-hospital mortality of patients hospitalized with a confirmed diagnosis of COVID-19 was evaluated as an outcome variable. Mortality after hospital discharge was not considered. These data were obtained by reviewing the virtual medical records of patients hospitalized from 14 May to 25 September 2020.

#### 2.3.2. Exposure Variable: ALRI

The ALRI was calculated using the following: ALRI = aspartate aminotransferase (U/L)/lymphocyte [14]. 

##### Laboratory Biomarkers

APRI was calculated as AST (U/L)/upper limit of normal value (U/L)/platelet count × 100 [11]. AST-to-neutrophil ratio index (ANRI) was calculated by dividing the AST (U/L) by the neutrophil count [10]. Neutrophil–lymphocyte ratio (NLR) was obtained by dividing the total absolute neutrophil counts over the total lymphocyte counts [15]. Platelets–lymphocyte ratio (PLR) was determined using the following formula: PLR = absolute platelet count/absolute lymphocyte count [16]. SII = platelet count × neutrophil count/lymphocyte count (×10^9^/L) [8]. The leukocyte glucose index (LGI) was defined as the product between blood leukocytes counts and glucose levels divided by 1000 [17]. Lactate dehydrogenase (LDH)/lymphocyte ratio was obtained as the ratio between LDH levels (U/L) and lymphocyte counts (cells/µL) [9]. The first measure of laboratory markers during the first 24 h of hospital admission were considered for this study.

#### 2.3.3. Other Variables

Demographic characteristics such as age, sex, and comorbidities (diabetes, hypertension, obesity, chronic kidney disease (CKD), and chronic obstructive pulmonary disease (COPD)) were collected. The laboratory parameters measured in the first 24 h of hospital admission included the following: hemoglobin (g/dL), hematocrit (%), leukocytes (×10^9^/L), platelets (×10^9^/L), neutrophils (×10^9^/L), lymphocytes (×10^9^/L), glucose (mg/dL), urea (mg/dL), creatinine (mg/dL), LDH (U/L), bilirubin (mg/dL), liver enzymes (AST and alanine aminotransferase (ALT)), and electrolytes.

### 2.4. Statistical Analysis

Qualitative variables are described as numbers or percentages, and quantitative data are expressed as mean (±standard deviation, SD) for normally distributed and median (interquartile range, IQR) for non-normally distributed. The distribution of quantitative variables was assessed using the Shapiro–Wilk test. The Mann–Whitney test or Student’s-t test was used to compare the quantitative variables between the different groups. The Chi-square test and Fisher’s exact test were used to evaluate the differences between the qualitative variables between the groups. The Receiver Operator Characteristic (ROC) curve [18] was calculated to find the accuracy of the variables in predicting COVID-19 mortality. An area under the curve (AUC) values were considered adequate at 0.7 to 0.8. Youden’s J index finds the cut-off value that maximizes the sum of sensitivity and specificity (or equally minimizes the sum of false positive and false negative errors) and is calculated as J = (sensitivity + specificity-1) [19]. The lymphocytes, PTT, AST, ALRI, NLR, SII, and LDH/lymphocyte ratio levels were dichotomized by defining the best cut-off obtained for Youden’s-J statistic on the ROC curve. Survival curves were estimated using the Kaplan–Meier method [20], and the differences between groups were assessed using the log-rank test [21]. Univariate and multivariate Cox proportional hazard regression models were adopted to calculate the hazard ratio (HR) with a corresponding 95% confidence interval (CI) [22] for mortality risk. The covariates that were significant in the univariate Cox regression analysis were selected to be tested in the multivariate Cox regression analysis. The variance inflation factors (VIF) were calculated to evaluate multicollinearity. The statistical assumptions for the regression analysis were met when there was no multicollinearity. VIF > 5 was used to identify highly correlated variables. None of the variables included in the model had a VIF > 1.5, thus indicating that there were no issues with multicollinearity, i.e., no multicollinearity between the independent variables was found. All analyses were 2-tailed, and a *p*-value < 0.05 was considered statistically significant. *p*-values were adjusted using Bonferroni correction (*p_corrected_*) to compensate for the effect of multiple hypothesis testing. The variables were filtered using <0.05 as a significance cut-off. Data analysis was performed using SPSS Statistics v.25 (SPSS Inc., Chicago, IL, USA), R v4.03 Statistical Software (R Foundation, Vienna, Austria), and MedCalc Statistical Software (MedCalc Software Ltd., Ostend, Belgium).

## 3. Results

### 3.1. Clinical Characteristics of the Study Subjects

The current study included 225 patients admitted to hospital with COVID-19, including 144 surviving patients and 81 non-surviving patients. The mean age of the non-survival group was 65 ± 12.705, and that of the survival group was 60.2 ± 13.515 (*p* = 0.004).The results of the laboratory tests at the time of hospitalization are summarized in Table 1. In the non-survival group, the urea, AST, ALRI, APRI, NLR, PLR, SII, and LDH/lymphocyte ratio were significantly higher than in the survival group (*p* < 0.05), while the lymphocyte count and partial thromboplastin time (PTT) values were significantly lower than that of the survival group (*p* < 0.0001). However, the age, urea, APRI, and PLR did not remain significant when multiple hypothesis correction was performed (*p_corrected_* ≥ 0.05).

### 3.2. Use of the Best Cut-Off Values of Hematologic Parameters to Discern COVID-19 Mortality

The best cut-off values of the lymphocytes, PTT, AST, ALRI, NLR, SII, and LDH/lymphocyte ratio were calculated using ROC analysis to identify the non-surviving patients. The area under the curve (AUC) values regarding lymphocytes, PTT, and AST were 0.67 (SE = 0.03; 95% CI: 0.6–0.73), 0.64 (SE = 0.03; 95% CI: 0.57–0.7), and 0.64 (SE = 0.03; 95% CI: 0.58–0.71), respectively (Figure 1A). Regarding ratios, the AUC of ALRI, NLR, SII, and LDH/lymphocyte ratio were 0.81 (SE = 0.02; 95% CI: 0.76–0.86), 0.67 (SE = 0.03; 95% CI: 0.6–0.73), 0.63 (SE = 0.03; 95% CI: 0.57–0.69), and 0.68 (SE = 0.03; 95% CI: 0.61–0.74), respectively (Figure 1B).

Lymphocytes, PTT, and AST could not be used in this study as potential diagnostic tools for subsequent analysis because their sensitivity or specificity was <50% (i.e., no better than chance) (Table 2). The best cut-off values for lymphocytes (≤1.21), PTT (≤31.9), AST (>26), ALRI (>42.42), NLR (>9.25), SII (>1857), and LDH/lymphocyte ratio (>0.369) were determined using Youden’s J index. 

### 3.3. Analysis of the Association of Hematologic and Enzymatic Parameters with COVID-19 Mortality

To identify the risk factors related to COVID-19 mortality, the seven statistically significant variables (*p_corrected_* < 0.05) in Table 1 were included in our analysis by using Kaplan–Meier curves and the univariate Cox regression model. The Kaplan–Meier curves were created using the best cut-off points obtained by the Youden’s J index for each variable. The mean survival time for the COVID-19 patients above the stated lymphocyte, PTT, AST, ALRI, NLR, SII, and LDH/lymphocyte ratio best cut-off values were 11.89 (SE = 0.71), 12.82 (SE = 0.81), 9.94 (SE = 0.83), 7.8 (SE = 0.86), 8.31 (SE = 0.78), 8.92 (SE = 0.76), and 7.9 (SE = 0.65) days, respectively. In comparison, the mean survival time for COVID-19 patients with below the stated lymphocyte, PTT, AST, ALRI, NLR, SII, and LDH/lymphocyte ratio best cut-off values were 8.05 (SE = 0.5), 9.58 (SE = 0.87), 11.57 (SE = 0.87), 11.69 (SE = 0.59), 13.66 (SE = 1.02), 13.62 (SE = 1.18), and 13.53 (SE = 1.21) days, respectively (Figure 2A–G). The differences in survival for patients with laboratory and enzymatic parameters above the stated best cut-off point compared to those below the cut-off point were statistically significant (*p* < 0.05).

Regarding the univariate Cox regression model elaborated with the seven variables that had statistically significant differences when multiple hypotheses correction was performed (*p_corrected_* < 0.05), lower levels of lymphocyte (HR = 2.69, 95% CI: 1.48–4.88; *p_corrected_* = 0.007) and PTT (HR = 2.68, 95% CI: 1.45–4.94; *p_corrected_* = 0.007) and higher levels of ALRI (HR = 3.2, 95% CI: 1.98–5.17; *p_corrected_* < 0.0001), NLR (HR = 2.01, 95% CI: 1.28–3.16; *p_corrected_* = 0.014), and LDH/lymphocyte ratio (HR = 2.3, 95% CI: 1.43–3.69; *p_corrected_* = 0.004) were predictors of in-hospital mortality (Table 3). Nevertheless, multivariate Cox regression reveled that ALRI [>42.42] (HR = 2.32, 95% CI: 1.35–3.97; *p_corrected_* = 0.01) was a prognostic factor of COVID-19 mortality (Table 3).

### 3.4. Demographic, Clinical, and Hematological Characteristics of COVID-19 Patients Stratified According to ALRI Values

Patients with lower ALRI levels were more frequently diabetic than those patients with high ALRI levels (58.3% vs. 41.9%, *p* = 0.015). However, diabetes did not remain significant when multiple hypothesis correction was performed (*p_corrected_* = 0.615). Patients with higher ALRI values had increased mortality (61.3% vs. 18.2%, *p_corrected_* < 0.0001) (Table 4). Moreover, patients with ALRI >42.42 had less lymphocytes (*p_corrected_* < 0.0001) and lower PTT (*p_corrected_* = 0.004) when compared with patients with ALRI ≤ 42.42, as well as increased values of AST (*p_corrected_* < 0.0001), ALT (*p_corrected_* = 0.0003), chloride (*p_corrected_* = 0.041), APRI (*p_corrected_* = 0.0001), ANRI (*p_corrected_* = 0.012), NLR (*p_corrected_* < 0.0001), PLR (*p_corrected_* = 0.0004), SII (*p_corrected_* = 0.0001), and LDH/lymphocyte ratio (*p_corrected_* = 0.0001) (Table 4). 

### 3.5. Correlation between Hematological Parameters and ALRI Levels at Admission in COVID-19 Patients

Spearman’s rank correlation coefficient was used to determine the correlation between ALRI and the hematologic parameters that had a significant difference in the comparison of patients with low and high ALRI levels. Most of the variables showed a weak correlation (value of r = −0.3 to −0.1 or 0.1 to 0.3). A statistically significant negative correlation was observed between lymphocytes (r = −0.514 [95% IC −0.604 to −0.410]; *p* < 0.0001), PTT (r = −0.265 [95% IC −0.383 to −0.139]; *p* = 0.0001), and chloride (r = −0.206 [95% IC −0.328 to −0.077]; *p* = 0.001). ALRI had a moderate correlation (value of r = 0.3 to 0.5) with ALT, APRI, NLR, PLR, and SII. Also, ALRI had a strong correlation with LDH/lymphocyte ratio (r = 0.547 [95% IC 0.448 to 0.633]; *p* < 0.0001) (Figure 3).

## 4. Discussion

This study reported data from 225 patients consecutively admitted with COVID-19 at one tertiary hospital in Mexico. Of these, 81 (36%) patients did not survive during hospitalization. Dessie and Zewotir suggested that the mortality rate of COVID-19 ranges from 3.14 to 61.51% [3]. Demographic data show that non-surviving COVID-19 patients are significantly older than COVID-19 survivors; in a previous report, COVID-19 mortality was significantly related to age, with approximate rates of 40% at the age 70+ years [23]. We found a mean age of 65 ± 12.705 in the non-surviving patients. However, no statistically significant differences were observed. Although comorbidities such as diabetes, hypertension, chronic kidney disease (CKD), chronic obstructive pulmonary disease (COPD), and cardiovascular disease each increased the risk of mortality [24], in our study, these variables were not associated with COVID-19 mortality. 

On the other hand, a decrease in blood platelet count and an increase in leukocytes and neutrophils have been associated with mortality in patients with COVID-19 [5,6]. Our results showed no significant differences in the mean or median regarding the platelet, leukocyte, and neutrophil concentrations of the non-surviving and surviving patients. In contrast, the median value of lymphocytes was significantly lower in non-surviving than in surviving patients, as previously reported [6]. PTT and AST had statistical significance between groups (*p_corrected_* < 0.05), as reported by Mahdavi et al., Hodges et al., and Ding et al. [12,25,26]. However, this study could not use lymphocytes, PTT, and AST as potential diagnostic tools because their sensitivity or specificity was <50% (i.e., no better than chance) [27]. Other reports have identified NLR (>9.1) [7], PLR (356.6) [28], SII (1835) [8], LDH/lymphocyte ratio (0.21) [9] as independent risk factors for COVID-19 mortality. Our study’s best cut-off points for NLR, SII, and LDH/lymphocyte ratio were >9.25, >362.5, >1857, and >0.369, respectively. NLR, SII, and LDH/lymphocyte ratio were not predictors of mortality despite a statistically significant AUC. Interestingly, PLR was also not a predictor of mortality in this study. These results on PLR and SII in COVID-19 mortality agree with that of previous reports [7,29]. Some studies that have reported an association between PLR and COVID-19 mortality included patients with thrombocytopenia, generating a confusion variable [4,30]. Furthermore, the results of a meta-analysis performed by Sarker et al. suggest that the prognostic value of PLR is heterogeneous and low-quality evidence [31]. 

Regarding liver enzyme-derived ratios, ANRI and APRI had no statistically significant difference between the study groups (*p_corrected_* ≥ 0.05). ANRI and APRI were also excluded from the univariate Cox regression analysis. Therefore, the liver enzyme-derived ratios analyzed in this study show little relevance (ANRI and APRI), except ALRI. Interestingly, APRI > 0.64 on admission was associated with severe COVID-19, whereas ALRI > 24.1 was not accepted as an independent risk factor for severe cases of COVID-19 [11]. In contrast, we found that ALRI > 42.42 could predict COVID-19 mortality. Hence, the role of ALRI as a predictor may depend on disease progression.

NLR, LDH/lymphocyte ratio, and ALRI showed a superior prognostic possibility for mortality with specificity and sensitivity > 60%. Kaplan–Meier survival curves obtained with best cut-off values obtained from ROC curves demonstrated that mortality was significantly associated with lymphocytes, PTT, AST, ALRI, NLR, SII, and LDH/lymphocyte ratio. Nevertheless, only the ALRI remained associated with mortality after multivariate Cox regression analysis. 

ALRI includes one liver enzyme (AST) and one peripheral blood parameter (lymphocytes), comprehensively summarizing the balance of liver inflammation (the organ most susceptible to SARS-CoV-2 infection [32]) and host immune response. ALRI (>18.734) is related to a poor prognosis in hepatocellular carcinoma patients after hepatectomy [13]. In our study, the ALRI values in non-surviving patients were significantly higher than that in the surviving patients. Additionally, ALRI > 42.42 was identified as an independent risk factor (HR = 2.32, 95% CI: 1.35–3.97; *p_corrected_* = 0.01) for COVID-19 mortality. Because there are no studies on the relationship between ALRI and COVID-19 mortality, we hypothesized that liver abnormalities in COVID-19 (association between early AST elevation and COVID-19 mortality), secondary to inflammation during SARS-CoV-2 infection [12] and lymphopenia (linked to the ability of SARS-CoV-2 to infect T cells [33]), could be an indicator of poor prognosis in COVID-19 patients [6] and suggest the relevance of ALRI increase in predicting patient mortality. Data from the COVID-19 Lombardy ICU Network in Italy show that age is the ultimate risk factor for COVID-19 mortality [34]. However, our study suggests that ALRI (>42.42) could be another risk factor for COVID-19 mortality compared to age (>61 years) upon admission in the Mexican population.

In our study, the patients with higher ALRI values were associated with increased values of ALT, chloride, APRI, ANRI, NLR, PLR, SII, and LDH/lymphocyte ratio. The association between these hematologic parameters and higher ALRI values requires further investigation. Interestingly, we found that ALRI levels showed significant positive moderate/strong correlations with ALT, APRI, NLR, PLR, SII, and LDH/lymphocyte ratio, indicating its potential value in evaluating COVID-19 mortality. Therefore, a combination of these inflammatory biomarkers may be beneficial for assessing and managing patients with COVID-19.

The present study has some limitations that need to be addressed. First, this study was a retrospective, single-center study with a relatively small sample size, which may affect the generalization of the results due to the enrollment limitations of enrolled patients. Nevertheless, these results could be compared to other studies with a similar number of patients to determine clinical predictors for COVID-19 outcomes [35,36,37]. The results of this study should be validated via comparison with other studies or future efforts focused on prospective analyses to strengthen our understanding of the utility of ALRI. The second limitation of this study is that most of the studies on NLR, PLR, LDH/lymphocyte ratio, and SII are retrospective and from Asian countries; thus, the comparisons made in this study may not have been strong enough. On the other hand, our study has some strengths, namely (a) it demonstrates the predictive prognostic value of ALRI independently of comorbidities and (b) to the best of our knowledge, this study is the first to determine the prognostic role of ALRI compared with other biomarkers in a COVID-19 cohort free of confounders affecting the count of hematologic parameters and liver enzymes. Furthermore, we applied Bonferroni correction, a stringent procedure, to discard the slightly significant variables. Although comparing the strength of different ratios with the same divisor (lymphocyte count) makes little sense, we decided to evaluate which of all these ratios is a better predictor of COVID-19 mortality.

## 5. Conclusions

In summary, our study suggests that ALRI at admission is a biomarker of mortality in COVID-19 patients. Although the cut-off value of ALRI is yet to be standardized, the prognostic role of ALRI in COVID-19 mortality is promising. Assessing this enzymatic/hematologic parameter may help to identify patients with a high risk of mortality. Early medical intervention for high-risk patients may reduce the fatality rate associated with this disease.

## Figures and Tables

**Figure 1 microorganisms-11-02894-f001:**
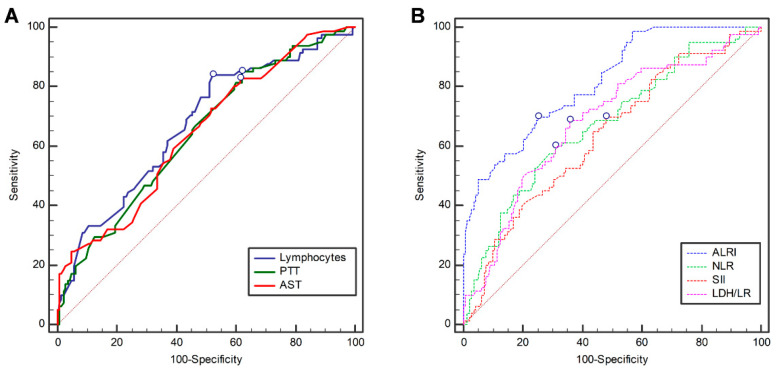
Receiver operating characteristic (ROC) curves of (**A**) lymphocytes (blue line), PPT (green line), AST (red line), and the ratios (**B**) ALRI (blue dotted line), NLR (green dotted line), SII (red dotted line), and (purple dotted line) LDH/LR in predicting mortality in patients with COVID-19. The blue circles indicate the best cut-off values. Abbreviations: PTT, partial thromboplastin time; AST, aspartate aminotransferase; ALRI, aspartate aminotransferase-to-lymphocyte ratio index; NLR, neutrophil-to-lymphocyte ratio; SII, systemic immune-inflammation index; LDH/LR, lactate dehydrogenase/lymphocyte ratio.

**Figure 2 microorganisms-11-02894-f002:**
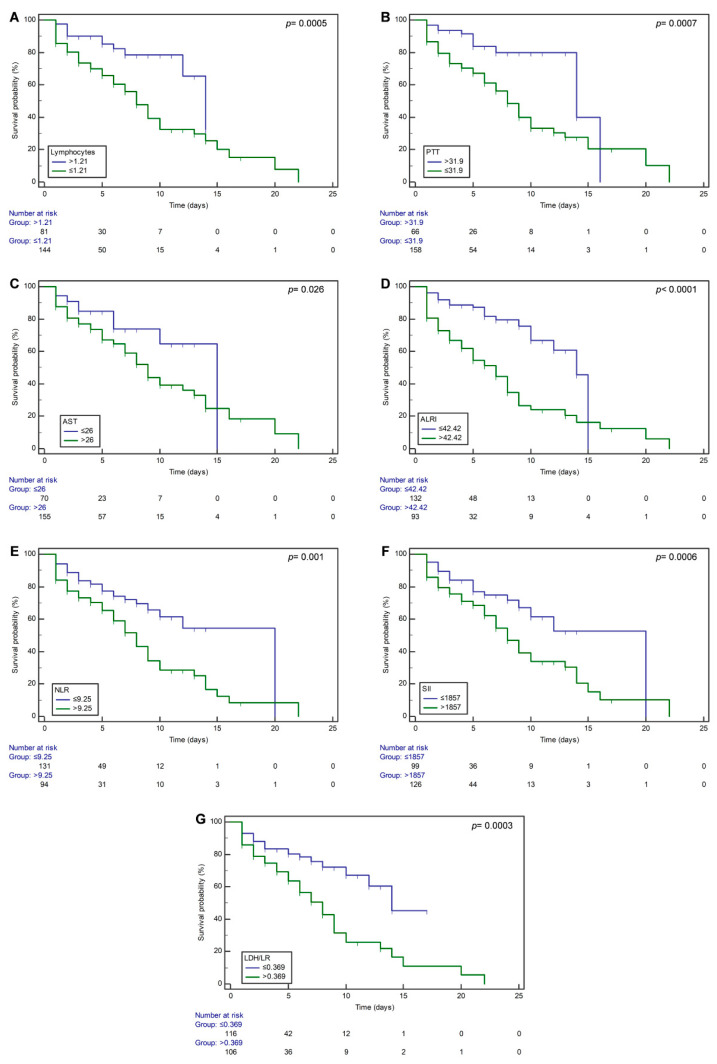
Kaplan–Meier survival curves of hospitalized COVID-19 patients according to the established best cut-off values of (**A**) lymphocytes, (**B**) PTT, (**C**) AST, (**D**) ALRI, (**E**) NLR, (**F**) SII, and (**G**) LDH/LR. A *p*-value <0.05 was considered statistically significant. The overall survival curves were estimated using the Kaplan–Meier method and *p* values (obtained via log-rank test). Abbreviations: PTT, partial thromboplastin time; AST, aspartate aminotransferase; ALRI, aspartate aminotransferase-to-lymphocyte ratio index; NLR, neutrophil-to-lymphocyte ratio; SII, systemic immune-inflammation index; LDH/LR, lactate dehydrogenase/lymphocyte ratio.

**Figure 3 microorganisms-11-02894-f003:**
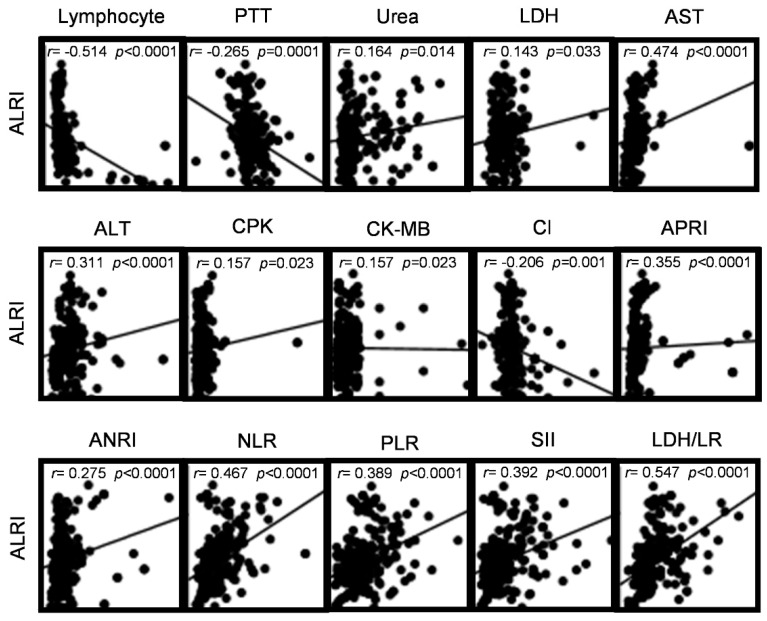
Spearman correlations between ALRI and hematologic parameters in COVID-19 patients. Scatter plots showing the correlation between ALRI and laboratory variables (data on admission). Weak correlation (value of r = −0.3 to −0.1 or 0.1 to 0.3); moderate correlation (value of r = −0.5 to −0.3 or 0.3 to 0.5); strong correlation (value of r = −1.0 to −0.5 or 0.5 to 1). Spearman’s rank correlation coefficient was used to evaluate the correlations. A *p*-value < 0.05 was considered statistically significant. Abbreviations: PTT, partial thromboplastin time; LDH, lactate dehydrogenase; AST, aspartate aminotransferase; ALT, alanine aminotransferase; CPK, creatinine phosphokinase; CK-MB, creatine kinase myocardial band; Cl, chloride; ALRI, aspartate aminotransferase-to-lymphocyte ratio index; APRI, aspartate aminotransferase-to-platelet ratio index; ANRI, aspartate aminotransferase-to-neutrophil ratio index; NLR, neutrophil-to-lymphocyte ratio; PLR, platelet-to-lymphocyte ratio; SII, systemic immune-inflammation index; LDH/LR, lactate dehydrogenase/lymphocyte ratio.

**Table 1 microorganisms-11-02894-t001:** Characteristics of the study population according to their COVID-19 survival.

Variable	Total (n = 225)	Survival Group (n = 144)	Non-Survival Group (n = 81)	*p*-Value	Adjusted *p*-Value
Sex, female	99 (44%)	68 (47.2%)	31 (38.3%)	0.194	1
Age (years old)	61.97 ± 13.39	60.2 ± 13.515	65 ± 12.705	0.004	0.168
Obesity	50 (22.2%)	30 (20.8%)	20 (24.7%)	0.504	1
Diabetes	116 (51.6%)	80 (55.6%)	36 (44.4%)	0.109	1
Hypertension	143 (63.6%)	87 (60.4%)	56 (69.1%)	0.192	1
CKD	45 (25%)	27 (18.8%)	18 (22.2%)	0.532	1
COPD	6 (2.7%)	3 (2.1%)	3 (3.7%)	0.670	1
Cardiovascular disease	11 (4.9%)	9 (6.3%)	2 (2.5%)	0.335	1
Days of hospitalization	4 (4)	4 (4)	3 (6)	0.206	1
Hemoglobin (g/dL)	12.8 (2.7)	12.95 (2.3)	12.3 (3.5)	0.865	1
Hematocrit (%)	39 (6.2)	39 (6)	39 (8.9)	0.738	1
Leukocytes (×10^9^/L)	10.2 (5.31)	10.29 (5.673)	10 (4.61)	0.615	1
Platelets (×10^9^/L)	300 (176)	304.623 ± 126.045	298.253 ± 117.407	0.601	1
Neutrophils (×10^9^/L)	7.23 (4.8)	7.145 (4.65)	8 (4.79)	0.140	1
Lymphocytes (×10^9^/L)	1 (0.720)	1.065 (0.893)	0.8 (0.5)	**<0.0001**	**0.0009**
PT (s)	13 (3)	13 (3.2)	13 (2.9)	0.696	1
PTT (s)	29 (5)	30 (6)	28.3 (4.8)	**<0.0005**	**0.014**
INR	1.2 (0.2)	1.2 (0.2)	1.2 (0.2)	0.554	1
Glucose (mg/dL)	130 (86)	126 (83)	130 (82)	0.248	1
Urea (mg/dL)	40 (54)	40 (36.3)	50 (94)	0.014	0.588
Creatinine (mg/dL)	0.9 (1.1)	0.9 (0.8)	1 (1.7)	0.576	1
LDH (U/L)	360 (213)	363 (198)	351 (254)	0.438	1
TB (mg/dL)	0.9 (0.5)	0.9 (0.4)	0.8 (0.6)	0.753	1
DB (mg/dL)	0.4 (0.3)	0.4 (0.3)	0.4 (0.3)	0.908	1
IB (mg/dL)	0.5 (0.3)	0.5 (0.3)	0.5 (0.3)	0.564	1
AST (IU/L)	40 (28)	34.5 (26)	43 (30)	**<0.0005**	**0.009**
ALT (IU/L)	33 (26)	30.5 (31)	37 (22)	0.136	1
CPK (IU/L)	157 (240)	143 (184)	160 (308)	0.091	1
CK-MB (IU/L)	24 (28)	23.5 (26.8)	24 (32.2)	0.263	1
BNP (pg/mL)	56 (138)	56 (160.8)	50 (138)	0.870	1
MYO (ng/mL)	200 (176)	200 (179)	259 (162)	0.056	1
Na (mmol/L)	137 (5)	137 (5)	137 (5)	0.377	1
K (mmol/L)	4.5 (0.7)	4.5 (0.7)	4.5 (0.7)	0.595	1
Cl (mmol/L)	99 (3)	99 (3)	99 (2)	0.678	1
ALRI	37.31 (25.274)	31.428 (25.773)	51.136 (36.6)	**<0.0001**	**<0.0001**
APRI	0.323 (0.29)	0.284 (0.26)	0.403 (0.41)	0.001	0.050
ANRI	4.777 (4.06)	4.6 (3.88)	4.78 (5.59)	0.127	1
NLR	8.315 (7.669)	7.481 (6.731)	10.302 (9.884)	**<0.0001**	**0.0007**
PLR	302.727 (286.31)	278.4 (256.31)	361.428 (281)	0.003	0.126
SII	2133 (2482)	1838 (2367.131)	2890 (3568)	**0.001**	**0.042**
LGI	73.75 (63.959)	72.894 (65.619)	76.9 (57.494)	0.721	1
LDH/LR	0.361 (0.304)	0.331 (0.271)	0.5 (0.333)	**<0.0001**	**<0.0001**

Data are presented as mean ± standard deviation, median (interquartile range), n (%). *p* values were calculated using Student’s-*t* test, Mann–Whitney U test, Chi-squared test, or Fisher’s exact test as appropriate. The bold values pertain to statistical significance (*p_corrected_* < 0.05). Abbreviations: CKD, chronic kidney disease; COPD, chronic obstructive pulmonary disease; PT, prothrombin time; PTT, partial thromboplastin time; INR, international normalized ratio; LDH, lactate dehydrogenase; TB, total bilirubin; DB, direct bilirubin; IB, Indirect bilirubin; AST, aspartate aminotransferase; ALT, alanine aminotransferase; CPK, creatinine phosphokinase; CK-MB, creatine kinase myocardial band; BNP, B-type natriuretic peptide; MYO, myoglobin; Na, sodium; K, potassium; Cl, chloride; ALRI, aspartate aminotransferase-to-lymphocyte ratio index; APRI, aspartate aminotransferase-to-platelet ratio index; ANRI, aspartate aminotransferase-to-neutrophil ratio index; NLR, neutrophil-to-lymphocyte ratio; PLR, platelet-to-lymphocyte ratio; SII, systemic immune-inflammation index; LGI, leukocyte glucose index; LDH/LR, lactate dehydrogenase/lymphocyte ratio.

**Table 2 microorganisms-11-02894-t002:** Receiver operating characteristic (ROC) curves, best cut-off points, and prognostic accuracy of hematologic and enzymatic parameters in COVID-19 mortality.

Variable	AUC	95% CI	*p*-Value	Best Cut-Off Point	Sensitivity (%)	Specificity (%)
Lymphocytes	0.67	0.6–0.73	**<0.0001**	≤1.21	83.95	47.22
PTT	0.64	0.57–0.7	**0.0001**	≤31.9	85.19	37.76
AST	0.64	0.58–0.71	**0.0001**	>26	82.72	38.89
ALRI	0.81	0.76–0.86	**<0.0001**	>42.42	70.37	75
NLR	0.67	0.6–0.73	**<0.0001**	>9.25	60.49	68.75
SII	0.63	0.57–0.69	**0.0004**	>1857	70.37	52.08
LDH/LR	0.68	0.61–0.74	**<0.0001**	>0.369	68.75	64

The bold values pertain to statistical significance (*p* < 0.05). Abbreviations: PTT, partial thromboplastin time; AST, aspartate aminotransferase; ALRI, aspartate aminotransferase-to-lymphocyte ratio index; NLR, neutrophil-to-lymphocyte ratio; SII, systemic immune-inflammation index; LDH/LR, lactate dehydrogenase/lymphocyte ratio.

**Table 3 microorganisms-11-02894-t003:** Univariate and multivariate Cox regression analysis of the variables associated with COVID-19 mortality.

	Univariate				Multivariate			
Variable	HR	95% CI	*p*-Value	Adjusted *p*-Value	HR	95% CI	*p*-Value	Adjusted *p*-Value
Lymphocytes	2.69	1.48–4.88	**0.001**	**0.007**	1.2	0.56–2.54	0.627	1
PTT	2.68	1.45–4.94	**0.001**	**0.007**	2.08	1.08–4.01	0.028	0.14
AST	1.86	1.04–3.32	0.035	0.245	-	-	-	-
ALRI	3.2	1.98–5.17	**<0.0001**	**<0.0001**	2.32	1.35–3.97	**0.002**	**0.01**
NLR	2.01	1.28–3.16	**0.002**	**0.014**	1.31	0.8–2.15	0.275	1
SII	1.88	1.16–3.03	0.009	0.063	-	-	-	-
LDH/LR	2.3	1.43–3.69	**0.0006**	**0.004**	1.2	0.66–2.2	0.538	1

Univariate Cox regression analysis was performed using the seven variables (lymphocyte, PTT, AST, ALRI, NLR, SII, and LDH/lymphocyte ratio) that had statistically significant differences when multiple hypotheses correction was performed (*p_corrected_* < 0.05). Candidate predictors with statistically significant differences (*p_corrected_* < 0.05) in the univariate Cox regression analysis were included in our multivariate Cox regression analysis. Hazard ratios (HRs) and 95% Confidence Intervals (CI 95%) are reported. Variance inflation factors were computed for the final model to evaluate multicollinearity. The bold values pertain to statistical significance (*p_corrected_* < 0.05). Abbreviations: PTT, partial thromboplastin time; AST, aspartate aminotransferase; ALRI, aspartate aminotransferase-to-lymphocyte ratio index; NLR, neutrophil-to-lymphocyte ratio; SII, systemic immune-inflammation index; LDH/LR, lactate dehydrogenase/lymphocyte ratio.

**Table 4 microorganisms-11-02894-t004:** Demographic, clinical, and laboratory characteristics of COVID-19 patients stratified by the ALRI.

Variable	ALRI ≤ 42.42 (n = 132)	ALRI > 42.42 (n = 93)	*p*-Value	Adjusted *p*-Value
Sex, female	59 (44.7%)	40 (43%)	0.802	1
Age (years old)	61.11 ± 13.39	63.14 ± 13.38	0.096	1
Obesity	25 (18.9%)	25 (26.9%)	0.158	1
Diabetes	77 (58.3%)	39 (41.9%)	0.015	0.615
Hypertension	88 (66.7%)	55 (59.1%)	0.248	1
CKD	26 (19.7%)	19 (20.4)	0.892	1
COPD	3 (2.3%)	3 (3.2%)	0.693	1
Cardiovascular disease	7 (5.3%)	4 (4.3%)	1	1
Non-survival	24 (18.2%)	57 (61.3%)	**<0.0001**	**<0.0001**
Hemoglobin (g/dL)	12.7 (3.1)	12.85 (2.7)	0.794	1
Hematocrit (%)	39 (6.6)	39 (5.8)	0.802	1
Leukocytes (×10^9^/L)	10.36 (5.63)	9.6 (5.18)	0.797	1
Platelets (×10^9^/L)	296 (158.5)	300 (177.75)	0.990	1
Neutrophils (×10^9^/L)	7.15 (4.82)	7.58 (4.79)	0.093	1
Lymphocytes (×10^9^/L)	1.21 (1.04)	0.79 (0.42)	**<0.0001**	**<0.0001**
PT (s)	13 (3)	13 (3)	0.960	1
PTT (s)	30 (6)	28.15 (5)	**0.0001**	**0.004**
INR	1.2 (0.2)	1.2 (0.2)	0.901	1
Glucose (mg/dL)	130 (100)	125.5 (68)	0.588	1
Urea (mg/dL)	40 (44)	42 (94)	0.040	1
Creatinine (mg/dL)	0.9 (0.6)	1.05 (1.8)	0.114	1
LDH (U/L)	347 (180)	400 (292)	0.003	0.123
TB (mg/dL)	0.8 (0.5)	0.95 (0.6)	0.154	1
DB (mg/dL)	0.4 (0.2)	0.4 (0.3)	0.347	1
IB (mg/dL)	0.5 (0.3)	0.5 (0.3)	0.293	1
AST (IU/L)	28 (23)	45 (25)	**<0.0001**	**<0.0001**
ALT (IU/L)	24 (26)	40 (21)	**<0.0001**	**0.0003**
CPK (IU/L)	127 (160)	178.5 (284)	0.009	0.369
CK-MB (IU/L)	20 (23)	29 (32.3)	0.007	0.287
BNP (pg/mL)	50 (92.5)	67 (150)	0.212	1
MYO (ng/mL)	200 (162)	236 (200)	0.200	1
Na (mmol/L)	138 (6)	137 (5)	0.072	1
K (mmol/L)	4.5 (0.7)	4.5 (0.51)	0.878	1
Cl (mmol/L)	99 (2)	100 (3)	**0.001**	**0.041**
APRI	0.25 (0.26)	0.41 (0.36)	**<0.0001**	**0.0001**
ANRI	4.09 (4.2)	5.53 (4.17)	**0.0003**	**0.012**
NLR	6 (6.51)	10.38 (8.7)	**<0.0001**	**<0.0001**
PLR	252.63 (255.02)	343.57 (270.8)	**<0.0001**	**0.0004**
SII	1730.11 (2489)	3030.62 (3333.93)	**<0.0001**	**<0.0001**
LGI	76.05 (84.11)	72.35 (44.17)	0.720	1
LDH/LR	0.3 (0.297)	0.517 (0.356)	**<0.0001**	**<0.0001**

Data are presented as mean ± standard deviation, median (interquartile range), n (%). *p*-values were calculated using Student’s-*t* test, the Mann–Whitney U test, Chi-squared test, or Fisher’s exact test as appropriate. The bold values pertain to statistical significance (*p_corrected_* < 0.05). Abbreviations: CKD, chronic kidney disease; COPD, chronic obstructive pulmonary disease; PT, prothrombin time; PTT, partial thromboplastin time; INR, international normalized ratio; LDH, lactate dehydrogenase; TB, total bilirubin; DB, direct bilirubin; IB, indirect bilirubin; AST, aspartate aminotransferase; ALT, alanine aminotransferase; CPK, creatinine phosphokinase; CK-MB, creatine kinase myocardial band; BNP, B-type natriuretic peptide; MYO, myoglobin; Na, sodium; K, potassium; Cl, chloride; ALRI, aspartate aminotransferase-to-lymphocyte ratio index; APRI, aspartate aminotransferase-to-platelet ratio index; ANRI, aspartate aminotransferase-to-neutrophil ratio index; NLR, neutrophil-to-lymphocyte ratio; PLR, platelet-to-lymphocyte ratio; SII, systemic immune-inflammation index; LGI, leukocyte glucose index; LDH/LR, lactate dehydrogenase/lymphocyte ratio.

## Data Availability

Data supporting the reported results can be provided upon reasonable request by the corresponding author.

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
