# Peer review of "The Role of Aspartate Aminotransferase-to-Lymphocyte Ratio Index (ALRI) in Predicting Mortality in SARS-CoV-2 Infection"

_microorganisms, 2023, doi:10.3390/microorganisms11122894_

Round 1

Reviewer 1 Report

Greetings to the authors, I have read your paper and at leas for the first round of revision I have the following comments and questions:

Was the in hospital mortality due to respiratory failure due to COVID 19 or were patients with other causes of mortality and with additional SARS COV2 infection included ? This is not a huge patient lot and the details should be discussed. 

In the tables you should just write for example obesity instead of "obesity, yes" same for the others with "yes"

How did the authors choose the best cut off value of ALRI of 42.42 ? are there any standards for these values ?

Lines 365 to 367 so basically the authors state that the ratios that have been analyzed show little relevance ?

Lines 384 to 387 you mention that ALRI values ere higher in surviving patients and afterwards you state that ALRI was an independent risk factor for mortality then going on to say that there is no relationship between ALRI and covid mortality. This phrase is very confusing and again induces doubts in regards to the research. 

No major language faults. 

Author Response

September 22, 2023

Dear Editor-in-Chief

Microorganisms, MDPI

Dear Editor,

Enclosed, please find the revised version of the manuscript entitled: “The role of aspartate aminotransferase-to-lymphocyte ratio in-dex (ALRI) in predicting mortality in SARS-CoV-2 infection[microorganisms-2623407], which we are submitting for consideration for publication in Microorganisms.

We attended to all the suggestions of the Reviewers and performed additional modifications to complete this work.

RESPONSE TO REVIEWER'S COMMENTS

Reviewer 1

  1. Greetings to the authors, I have read your paper and at leas for the first round of revision I have the following comments and questions:

Reply:Thank you very much for your review and comments.

  1. Was the in hospital mortality due to respiratory failure due to COVID 19 or were patients with other causes of mortality and with additional SARS COV2 infection included ? This is not a huge patient lot and the details should be discussed.

Reply:This study only included patients with in-hospital mortality due to COVID-19. The patients with other causes of mortality and with additional SARS-CoV-2 infection included were excluded from study. This response was included in the Material and Methods Section (Page 2, Lines 79-81):

“This study only included patients with in-hospital mortality due to COVID-19. The patients with other causes of mortality and with additional SARS-CoV-2 infection included were excluded from study.”

On the other hand, we agree with the reviewer that this is not a huge patient lot. Thus, the details were discussed in this new submission, as follows (Page 14, Lines 403-409):

“The present study has some limitations that need to be addressed. First, this was retrospective, single-center, and had a relatively small sample size, which may affect the generalization of the results due to the enrollment limitation of enrolled patients. Nevertheless, these results could be compared to other studies with a similar number of patients performed to determine clinical predictors for COVID-19 outcomes [1-3]. The results of this study should be validated with other studies or future efforts focused on the prospective analyses to strengthen our understanding of the utility of ALRI.”

  1. In the tables you should just write for example obesity instead of "obesity, yes" same for the others with "yes"

Reply:The tables have been modified according to the reviewer's suggestion in this new submission. The word “yes” was removed from the tables.

  1. How did the authors choose the best cut off value of ALRI of 42.42 ? are there any standards for these values ?

Reply:The best cut off value of ALRI >42.42 was determined using the Youden´s J index, which is used to find the best cut off value that maximizes the sum of sensitivity and specificity (or equally minimizes the sum of false positive and false negative errors) and is calculated as J= (sensitivity + specificity-1). This response was included in Material and Methods Section (Page 3, Lines 135-138) and Results Section (Page 6, Lines 202-204), as follows:

“Youden´s J index finds the cut-off value that maximizes the sum of sensitivity and specificity (or equally minimizes the sum of false positive and false negative errors) and is calculated as J= (sensitivity + specificity-1) [4].”

“The best cut-off values for lymphocytes (1.21), PTT (31.9), AST (>26), ALRI (>42.42), NLR (>9.25), SII (>1857), and LDH/lymphocyte ratio (>0.369) were determined using the Youden´s J index.”

  1. Lines 365 to 367 so basically the authors state that the ratios that have been analyzed show little relevance ?

Reply: This is correct,the liver enzyme-derived ratios analyzed in this study show little relevance (ANRI and APRI), except ALRI. This response was included in the Discussion Paper (Page 13, Lines 361-364), as follows:

“Regarding liver enzymes-derived ratios, ANRI and APRI had no statistically significant difference between the study groups (pcorrected0.05). ANRI and APRI were also excluded from the univariate Cox regression analysis. Therefore, the liver enzyme-derived ratios analyzed in this study show little relevance (ANRI and APRI), except ALRI.”

  1. Lines 384 to 387 you mention that ALRI values ere higher in surviving patients and afterwards you state that ALRI was an independent risk factor for mortality then going on to say that there is no relationship between ALRI and covid mortality. This phrase is very confusing and again induces doubts in regards to the research.

Reply:This was a typographical error and was corrected in this new submission (Page 14, Lines 387-395).

The sentence “In our study, the ALRI values in surviving patients were significantly higher than in the non-surviving patients. Additionally, ALRI >42.42 was identified as an independent risk factor (HR= 2.76; p= 0.0003) for COVID-19 mortality. There is no study on the relationship between ALRI and COVID-19 mortality.” was changed to “In our study, the ALRI values in non-surviving patients were significantly higher than in the surviving patients. Additionally, ALRI >42.42 was identified as an independent risk factor (HR= 2.32, 95% CI: 1.35-3.97; pcorrected= 0.01) for COVID-19 mortality. Because there are no studies on the relationship between ALRI and COVID-19 mortality, we hypothesized that liver abnormalities in COVID-19 (association between early AST elevation and COVID-19 mortality) secondary to inflammation during SARS-CoV-2 infection [12], and lymphopenia (linked to the ability of SARS-CoV-2 to infect T cells [33]) as an indicator of poor prognosis in COVID-19 patients [6] might suggest the relevance of ALRI increase in predicting of patient mortality.”

  1. No major language faults.

Reply:Thank you very much for your review and comments.

Thank you very much for your comments. Please let me know about above questions.

By signing this letter, we acknowledge that all the authors participated sufficiently to take public responsibility for its content. All of the authors have given their consent for submission to the journal. Further, we have no commercial associations which impact this work.

Best regards,

Professor José Manuel Reyes, Ph.D.

Mexican Institute of Social Security (IMSS), Mexico

E-mail: jose.reyesr@imss.gob.mx; jmreyesrz@hotmail.com

References

  1. Gu, Y. et al. PaO2/FiO2 and IL-6 are risk factors of mortality for intensive care COVID-19 patients. Sci Rep 11, 7334 (2021).
  2. Song, C.-Y., Xu, J., He, J.-Q. & Lu, Y.-Q. Immune dysfunction following COVID-19, especially in severe patients. Sci Rep 10, 15838 (2020).
  3. Xu, J. et al. Associations of procalcitonin, C-reaction protein and neutrophil-to-lymphocyte ratio with mortality in hospitalized COVID-19 patients in China. Sci Rep 10, 15058 (2020).
  4. Youden WJ. Index for rating diagnostic tests. Cancer. 1950;3(1):32-35. doi:10.1002/1097-0142(1950)3

5          Wanner N, Andrieux G, Badia-i-Mompel P, et al. Molecular consequences of SARS-CoV-2 liver tropism. Nat Metab. 2022;4(3):310-319. doi:10.1038/s42255-022-00552-6

Reviewer 2 Report

Authors provided research about predictive value of different haematological parameters and their ratios in prediction of 14-day mortality from COVID-19.

Although concept could have sense, I have a few serious concerns. 

First of them would be the fact that you have chosen the 14 day period. All of us who were dealing with COVID-19 patients are aware of the fact that there is huge percentage of the patients who died during third or fourth week of hospitalisation. Thats why majority of the researchers choose 28-day period of follow up. So, choosing 14-day follow up I consider like bias.

Second, there are several biases in statistical methods. 

Why you have chosen 5 parameters for multivariate analysis, when you had 11 important parameters in univariate analysis?

Did you checked for multicolinearity before including them in analysis?

I am affreid that comparing strength of different ratios with the same divider (lymphocyte count) does not have a lot of sense...

Why Bonferoni correction for multiple comparison was not used?

How you can claim that some AUC ROC is bigger than other without statistical comparison of those values?

etc...

And third but not the least, where did you find the data that liver is target organ for SARS CoV 2 infection?

Author Response

September 22, 2023

Dear Editor-in-Chief

Microorganisms, MDPI

Dear Editor,

Enclosed, please find the revised version of the manuscript entitled: “The role of aspartate aminotransferase-to-lymphocyte ratio in-dex (ALRI) in predicting mortality in SARS-CoV-2 infection[microorganisms-2623407], which we are submitting for consideration for publication in Microorganisms.

We attended to all the suggestions of the Reviewers and performed additional modifications to complete this work.

RESPONSE TO REVIEWER'S COMMENTS

Reviewer 2

  1. Authors provided research about predictive value of different haematological parameters and their ratios in prediction of 14-day mortality from COVID-19.

Although concept could have sense, I have a few serious concerns.

First of them would be the fact that you have chosen the 14 day period. All of us who were dealing with COVID-19 patients are aware of the fact that there is huge percentage of the patients who died during third or fourth week of hospitalisation. Thats why majority of the researchers choose 28-day period of follow up. So, choosing 14-day follow up I consider like bias.

Reply:Thank you very much for your review and comments. We agree with the reviewer; many patients died during the third or fourth week of hospitalization. In this study, patients were followed up from admission to discharge (1 to 22 days) to analyze mortality risk. However, the Kaplan-Meier curve showed a cross-over at 14 days. Thus, we decided to show the behavior of the Kaplan-Meier curves at 14 days of follow-up as a reference parameter. In this new submission, the Kaplan-Meier curves show the 22-day follow-up of the patients (Figure 2A-G).

  1. Second, there are several biases in statistical methods.

Why you have chosen 5 parameters for multivariate analysis, when you had 11 important parameters in univariate analysis?

Reply:In this new submission, the univariate Cox regression analysis was performed using the 7 variables(lymphocyte, PTT, AST, ALRI, NLR, SII, and LDH/lymphocyte ratio) that had statistically significant difference when multiple hypotheses correction was performed (pcorrected<0.05). Candidate predictors with statistically significant difference (pcorrected<0.05) in univariate Cox re-gression analysis were included in multivariate Cox regression analysis. This response was included in the Results section of manuscript (Page 9:lines 246-248, and lines 260-264).

  1. Did you checked for multicolinearity before including them in analysis?

Reply:In this study, the variance inflation factors (VIF) were calculated to check multicollinearity. The statistical assumptions for the regression analysis were met when there was no multicollinearity. VIF >5 was used to identify highly correlated variables. None of the variables included in the model had a VIF >1.5, thus indicating no issues with multicollinearity. No multicollinearity between the independent variables was found. This response was included in the Material and Methods section (Page 3, Lines 146-150).

  1. I am affreid that comparing strength of different ratios with the same divider (lymphocyte count) does not have a lot of sense...

Reply:We agree with the reviewer that comparing the strength of different ratios with the same divisor (lymphocyte count) makes little sense. However, we decided to evaluate which of all these ratios sharing the same divisor (lymphocyte count) is a better predictor of COVID-19 mortality. This response was included in this new submission in Discussion Section (Page 14, Lines 418-420).

  1. Why Bonferoni correction for multiple comparison was not used?

Reply:In this new submission, p-values were adjusted using the Bonferroni corrections (pcorrected) to compensate the effect of multiple hypothesis testing. The variables were filtered using <0.05 as a significance cut-off. The Bonferroni correction is a stringent procedure to discard the slightly significant variables. This response was included in Material and Methods (Pages 3 and 4: Lines 151-153), and Discussion Section (Page 14, Lines 417-418).

  1. How you can claim that some AUC ROC is bigger than other without statistical comparison of those values?

etc...

Reply:We agree with the reviewer, we cannot claim that one AUC ROC is higher than another without statistically comparing those values. Thus, these statements were removed from the Abstract and Discussion (Page 13, Lines 370-371) Sectionin this new submission.

  1. And third but not the least, where did you find the data that liver is target organ for SARS CoV 2 infection?

Reply:We agree with the reviewer,Wanner et al. reported that the liver is an organ susceptible to SARS-CoV-2 infection (4), but they do not claim that it is a target organ for this coronavirus. Therefore, the statement: “AST-to-lymphocyte ratio index (ALRI) is a novel biomarker of survival in patients with hepato-cellular carcinoma, a target organ of SARS-CoV-2 infection” was modified to “AST-to-lymphocyte ratio index (ALRI) is a novel biomarker of survival in patients with hepato-cellular carcinoma, an organ susceptible to SARS-CoV-2 infection” (Abstract).

Thank you very much for your comments. Please let me know about above questions.

By signing this letter, we acknowledge that all the authors participated sufficiently to take public responsibility for its content. All of the authors have given their consent for submission to the journal. Further, we have no commercial associations which impact this work.

Best regards,

Professor José Manuel Reyes, Ph.D.

Mexican Institute of Social Security (IMSS), Mexico

E-mail: jose.reyesr@imss.gob.mx; jmreyesrz@hotmail.com
